# The Role of Cultural Factors in Sustainable Food Consumption—An Investigation of the Consumption Habits among International Students in Hungary

**Nikolett Nemeth [1], Ildiko Rudnak [2], Prespa Ymeri [1,*] and Csaba Fogarassy [3]** 

[1] Doctoral School of Management and Business Administration, Szent István University, 2010 Gödöllő, Hungary; nikolett.nemeth99@gmail.com

[2] Institute of Social Sciences and Teacher Training, Szent István University, 2100 Gödöllő, Hungary; Rudnak.Ildiko@gtk.szie.hu

[3] Climate Change Economics Research Centre, Szent István University, 2100 Gödöllő, Hungary; Fogarassy.Csaba@gtk.szie.hu

* Correspondence: Ymeri.Prespa@phd.uni-szie.hu

**Abstract:** Food consumption plays a pivotal role in the economy and the health of individuals. Foods and meals, in addition to sustaining life, also have many functions in society, such as human bonding. The purpose of our study is to present a qualitative research method to show the role of food consumption in the culture of several ethnic groups, and to introduce the ways in which cultural factors influence eating habits and local food supply conditions. In the first part of the research, the sample was a mix of multiple nationalities. During our investigations, the main questions were: What do you think about the culture and value food consumption? What kind of food do you consume the most? What differences do you find in the habits of different ethnic groups, especially regarding their eating habits? In the second part, we asked the main actors of the local supply system (restaurants, buffets, shops) about the ways they track the demand of foreign students. Our results have been implemented into two different SWOT matrixes. We can conclude that such research on food consumption attitudes and community behavior is essential. Most of the interviewed students are interested in comparing their diet and cultural traditions to those of other nations', and prefer local foods. The study proved that eating habits in Hungary have an impact on the eating habits of international students, and they changed them from several perspectives. The study found that dietary choices are complex decisions that have a significant environmental and social impact, but we need to add that thanks to the strong cultural background, the students can keep their sustainable eating and community values abroad, which can also strongly influence the development of the local food supply practices.

**Keywords:** cultural factors of consumption; sustainable diet; local food supply; sustainable food; healthy ingredients

---

## 1. Introduction

We agree that agriculture makes the highest contribution to the environmental burden. Thus, the choice of food types and diet is relevant in case of sustainable consumption, preparation and production, and other socio-economic factors like public health or social cohesion [1]. Between 2000 and 2050, livestock production will be doubled due to population growth and the increase in per capita meat consumption. The consumption of ruminants contributes significantly to the increase in land involved in agriculture. Meat production is not only contributing to greenhouse gas (GHG) emission due to modern animal breeding, but also via the production and fertilization of fodder crops [2]. The emphasis,

therefore, is on examining the causes of the environmental burden moving from population growth towards per capita consumption patterns [3]. Food consumer habits are very different today in developed and developing countries, but on the global market, the latter follows trends of the former. In response to rapidly changing demand, developing countries like China, India, and Brazil also face the problem of reorganizing supply chains that affect global flows for both human and animal consumption products. Advanced food companies have introduced their products to markets in developing countries to gain increased or growing income of consumers. The question once again is between demand and supply: what are the real food needs, how do cultural notions affect these processes in global systems? Can we maintain our sustainable eating values abroad? The liberalized trade and consumer aspirations are simultaneously changing forces of the evolution of dietary habits and food system [4]. Obesity and starvation at the global level are sources of many problems [5,6], while environmental pressure is growing due to overfishing, soil exploitation, water scarcity, the use of minerals for plant protection and stimulating chemicals, greenhouse gases [1,7], or because of food waste [8], which is enhanced by the energy use of food crops or farmland. Consumer confidence has been diminished, and experts are competing to win consumers, so a sustainable diet is a rather confusing term. The variability of food-related assumptions is maintained by the internet, the media and the globalized market [4]. In the meantime, in the background, there are the forces of exploitation and injustice, like malnutrition along with excessive food supply, unfavorable conditions of processing work on the supply side, or disproportions in benefits. Food security can thus be interpreted in several ways. It seems that today's giants have made goods a commodity, with all the accompanying positive and negative consequences [3,9].

Many questions have remained unanswered about the specifics and dynamics of change in sustainable consumption, which are not due to the shortcomings of previous research, but to the effects of rapid social change (globalization, migration). In our short study, we demonstrate the role played by the elements of culture in food consumption, what differences can be observed in the food consumption of different populations, and how cultural factors such as value, religion, traditions, ceremonies, etc. affect eating habits. According to our hypothesis, thanks to a strong cultural background, international students retain a substantial part of their nutritional and social values, even during their extended stay abroad. Our study is devoted to exploratory research, which provides a reasonable basis for later quantitative analysis.

The main differences between this paper and the current research are as follows: because international students retain their consumption habits during their extended stays abroad (six months or more), they have a significant influence on the dynamics of the consumption habits of their temporary home. By introducing cultural diets (e.g., Mediterranean—less meat, more vegetables), preferring healthier foods has an impact on the local food supply system. It can accelerate the trend of local food consumption and improve the appearance of healthy food in the food supply structure. These phenomena have not been observed so far in relation to the dynamics of change in sustainable consumption systems.

## 2. A Few Aspects of the Sustainability Dimension

Sustainability criteria for food consumption can be categorized with agricultural technology, which minimizes the use of soil-damaging substances and supports sustainable, renewable energy use. These include bio, biodynamic and integrated farms [10]. Freshwater-related eco-toxic research has shown that primary fodder wheat and other peas and cereals can have 18–91 times the negative externality effect on the environment, while soybeans can reach up to 1159 times higher compared to grass. For 1 kg of food produced, bread has twice, milk has three times, minced beef has 50 times, chicken filet has 138 times, and minced pork has 168 times higher externality values than the lowest effect of pea soup. The research clearly shows that the environmental impact of plant products is lower than that of animal products [11]. Besides, it is apparent from studies that the same amount of nutrients produced from plants produce 26–48% fewer pollutants, and have significantly less environmental

impacts than meat or dairy products [11–13]. This is also true in Germany. Here, in the case of agricultural land use, this difference is eightfold, while it is nine-fold in the acidification of the area, and fivefold for global warming [8]. However, in the case of crop production, the use of the appropriate method is desirable. Among the vegetable and fruit-growing technologies, the greenhouse has the most significant global warming potential (GWP), while there is hardly any difference between the unheated greenhouse or the open field cultivation. Beans contribute mainly to acidification of the soil, while leeks do the least. Investigations in the area, however, show that tomatoes are grown with the largest GWP, acidification and eutrophication, due to the intensive use of stimulants and pesticides [14].

It is a sad fact that due to the international food trade, the resources used for domestic products are utilized abroad [15]. In the case, for example, of Germany, the benefits of local freshwater used both directly and indirectly for producing food is mainly exported to Spain and Pakistan, and in the case of animal products, to France and Argentina. For land use, benefits primarily go to Argentina and Brazil, and through animal products to the Netherlands and the Czech Republic [8]. The availability potential of freshwater can be measured by the Water Allowance Coefficient (WAC), which can be expressed in financial terms after a monetary value adjustment, to make a more comprehensive and lighter comparison with natural results [16]. However, Du et al. 2017 [17] state that the water consumption behind food production displays a decreasing trend due to the decrease in meat consumption and an increase in fruit and vegetable consumption.

Sustainable eating is connected to different social (quality of life) areas, so its change can influence them. It is desirable to begin to understand this process in agriculture, as this is the basis of food production, and herein appear the most negative environmental externalities on the production side (structural and qualitative transformation of the soil due to the use of pesticides and stimulants, use of drinking water for irrigation purposes, water pollution, gaseous emissions, impurities from storage). Health is based on this, because the nutritional value of food depends on agricultural processes and input factors (if something is missing or in surplus, diseases come up) [18]. Concerning health, the availability of healthy food resources is a significant issue, which is more dependent upon socio-economic elements (employment, individual income, national welfare). This means that individual needs and information are not enough for proper nutrition, and legal-policy decisions must be supported. Nevertheless, we should not forget about the importance of taste, the presence of cultural customs and the influence of religion, which ultimately generates eating habits from generation to generation [19].

As agreed by governments, UN agencies and others, "Sustainable Diets are those diets with low environmental impacts which contribute to food and nutrition security and healthy life for present and future generations. Sustainable diets are protective and respectful of biodiversity and ecosystems, culturally acceptable, accessible, economically fair and affordable; nutritionally adequate, safe and healthy; while optimizing natural and human resources" [18]. Sustainable eating is therefore essential not only in terms of food and nutrient safety [20,21], but also on the social level [22,23]. That is why numerous recommendations and studies advise to policy makers in food security and sustainability regulation to integrate direct, indirect or voluntary tools into the social framework for sustainable food consumption [20,24–30].

Increasingly regional research provides a more comprehensive picture of the situation, mainly through the processing of care-side data. The result is also evident in figures: more local, seasonal and less energy-intensive foods have to be offered in institutions [31], as opposed to producing and supplying perishables to consumer's table in cities [32]. Recognizing this, the Food Distribution Planner has been developed, which considers carbon footprint, as well as shipping time and costs [33]. The increase of ruminants is associated with the highest GHG emissions and land use externalities. The impacts of poultry and plant products are the smallest, while beef and seafood have the most significant carbon footprint. For one kilogram of edible product, large stems and ruminant animals have by far the most significant carbon footprint. The most advantageous, from this point of view, is the production of shellfish, milk, poultry products, and plants.

Rank is also proven for protein. The most advantageous source of protein yields 150 times less carbon footprint than the least favorable one [34]. It has been conceived that in the production of animal products, a product that is unsuitable for human consumption can often be processed, from which the animal still produces milk or eggs for human consumption. On the other hand, there is also a kind of competition in the cultivation of edible and fodder crops. In less-favored regions, more impoverished people often sell their nutritional-rich animal products (dairy products and eggs) instead of consuming them themselves. However, in so doing, they endanger their food and nutrition security, but they also contribute to a more balanced and productive adult life of their children by spending more money on their education [35]. High consumption of meat makes social tension worse, and the problem of starvation, livestock farming plays a central role in sustainability policy. The results of the change are spectacular and ripple at the same time. The excessive consumption of meat is not only a burden to the environment, but it also leads to social downturn [36]. Food, environment, and health presuppose a relationship that is influenced by many factors, creating both challenges and opportunities [37]. A holistic approach is essential. Measurements should be applied and, if necessary, developed to balance changes in expenditure and budget. Redistribution should also contribute to the local economy by supporting alternative food systems. Plants, like grains, fruits, and vegetables have the lowest GWP value, while the meat of ruminants has the highest. Despite the barriers, it has been demonstrated that the proper use of appropriate methods can produce comprehensible comparisons for consumers so that the responsibility of their decisions can be made clear [7].

## 3. A Short Review of the Cultural Differences in Sustainable Food Consumption

When examining the economic factors affecting consumer behavior, primarily income and prices should be considered, as well as the structure of consumption expenditure. There are several ways in which income is affected by food consumption, i.e., on its size, its structure, its breadth, and depth. Initially, the food market was characterized by high production volumes at low prices, which pushed the quality down. Urban food consumption and food production have resulted in enormous changes in the environment. These days, due to the reorganization of the food economy and its impact on society, sustainability issues have also become evident in the food consumption side [4,38]. It is no longer a secret how food on the plate impacts the World's environmental, biodiversity and climate factors. Moreover, not merely the ingredients, but the production method is also crucial because high meat consumption requires intensive farming and animal husbandry methods, which is particularly unhealthy. It stands to reason, however, that meat consumption can be realized with smaller externalities [3]. The weekly contribution of an Australian household to global warming has been studied according to three diets. In the general diet, meat, fish and dairy products were included in the family menu (beef, lamb, whole chicken, chicken breast, eggs, tuna, fish, pork, ham, and salami). In the next diet, the ruminant meat was replaced (kangaroo instead of beef, rabbit, and duck instead of lamb, whole chicken, chicken breast, eggs, pollock instead of fish, pork, ham, and salami). In the third, meat was not included (peanuts, almonds, pinto beans, lentils, pilchards, muesli bars, peanut butter, baked beans, milk, cheese, yogurt, beans, breakfast cereals, pasta, white rice). After adjusting the level of consumption to the required volume of nutrition, it was found that choosing the second diet (free of ruminant meat) reduced dietary GWP by 30%, and choosing the meat-free third by 52% [7].

We can, therefore, see that diet choice is a complex decision with significant environmental and social impacts. Research studies have created a variety of diet groups in the course of researches and recommendations for ease of interpretation.

Beyond the minimum nutrition value, the individual's nutritional needs can be met in many ways. We can conclude that high-quality diets are realized when that intake of more of twenty vital nutrients, less of three harmful nutrients, and lower caloric intake than the average occur at the same time. The concept of food sustainability meets both the vital needs and the conditions of the pillars of sustainable development concurrently. The expression of a balanced diet is more common in public and, like in the Mediterranean diet, has no specific definition [39]. A sustainable diet is mainly

connected to the characteristics of people's diet living in Mediterranean countries. Most of these diets include olive oil, olives, fruits, vegetables, cereals, legumes, and nuts. Those contain fish and dairy products only in a smaller proportions, while meat or meat products are barely included [40]. As a further consideration of the Mediterranean diet, we can say that it is not just nutrition, but a kind of sustainable lifestyle. It is essentially nutrient-rich and rich in vitamins, contributes to the ideal body weight and reduces the risk of certain diseases (like metabolic syndrome, obesity, type-II diabetes) and aging. Doing all this by lowering the burden on the environment and enriching biodiversity enhances the socio-cultural value of food and stimulates the local economy [41]. In other guidelines, we can observe similar results. For example, according to Brazilian recommendations, meals should consist mainly of unprocessed or barely processed foods and avoid strongly processed ones at the same time. We should eat in company and share every duty regarding food preparation. Researchers and decision-makers have to promote recommendations tailored to and based on their circumstances, and make good, sustainable food consumption and customs, wherever they are in the world [26].

It has long been understood that the vegetarian diet (daily fresh fruit, raw salad, and whole wheat bread) contributes significantly less to fatal diseases such as ischemic heart disease, cerebro-vascular disease, malignant neoplasm, colorectal cancer, or breast cancer [42]. On the other hand, it is observed that the most popular sustainable eating habits among consumers comprise a meat-free day each week or smaller portions of meat. These are followed in the ranking by eating less, purchasing free-range meat, purchasing sustainable-labelled products, and buying organic vegetable and fruit. The purchase of bio-meat and bio-dairy products and lower consumption of dairy products were in the final positions [43,44]. Measures seeking to promote both the nutritional health and the sustainability aspects of food may interact to produce effects higher than those that would occur through uncoordinated action [29].

For sustainability, it does matter where we eat our food. Findings suggest that nutrition knowledge and sustainable mindsets have little influence on the eating decisions away from home. Changes in work and mobility patterns are very likely to have an impact on the way consumers eat away from home [45]. A Swiss study was made on how the food offered by a public supplier canteen contributes to the environmental burden. According to the results, the specific weight of agriculture in the supply chain is indisputable 58% of the total GWP, processing, and packaging respond for 12%, transport for 6%, while the operation of the canteen for the remaining 24%. From another aspect, meat products are responsible for 48% of the total GWP, durable foods for 13%, dairy products for 10% and vegetables for 8% [46]. By examining pasta dishes, it turned out that the method used in catering is also considered when serving the same meal. For example, the same pasta in a pasta maker requires 60% less power and 38% less water than on the range tops. It also matters whether you use electricity or gas or choose a cook-chill or cook-warm method. The use of electricity in both cases increases the environmental burden significantly by 13–98% (cook-chill) and 17–96% (cook-warm) [47]. From the data, it is apparent that the meals prepared and served in canteens or restaurants have higher energy consumption than home-made food, because of the use of more chilled or frozen raw materials and because these meals must be kept warm before serving. Also, in such kitchens, the vapor extractor works all day long. Preparation is the most energy-intensive stage for home-made food.

Moreover, research has also revealed that meat substitutes (Quorn$^{TM}$) are nutritionally richer than meat dishes [48]. The study of German diet habits also showed that eating animal products outside has a substantial environmental impact on the value chain, mainly because many of these products are lost somewhere on the supply chain. In the case of in-home consumed foods, this loss is 11–17%, while for out-of-home consumed meals it is 29 to 33%, depending on the environmental impact factor [8].

People's food habits are typically characterized by health, supply, consumer culture, family habits passed on from the previous generation, society, the environment and policy frameworks [1,4,28]. Consumer choice of food at present can be determined by whether the product is sustainable or organic. For example, most Italian consumers (69%) buy sustainable food at least occasionally, which may either be weekly or daily, while one-third of them (31.2%) rarely [10]. Nearly 64% of Italian households admit

to being environmentally conscious. Half of them consume a lot of fruit and vegetables, and they put great emphasis on purchasing sustainable food. They are also planning to do so in the future. Therefore, they mainly buy local or organic products on the market, or directly from the producer. The rest of the group is sensitive to the environment, but it was not so evident in their behavior, buying and consumption habits, as their choices were mainly determined by price [49,50]. Therefore, it is salutary that the meatless Italian diet has the lowest value of pollutants, energy consumption, and carbon footprint, and that it is also the cheapest for households together [51]; this is otherwise consistent with the result that the level of real income and content level of protein or calories of crops consumed are directly proportional, a relationship which has been observed since 1960 [52].

The results of our review are fully integrated into EU values. Life Cycle Analysis (LCA) helps to detect the environmental impact of dietary habits and their possible reduction by changing diet. In Europe, with the shift of 25–50% of the consumed nutrient from animal products (beef, dairy, pig meat, poultry, and eggs) to plant nutrition, significant effects can be achieved, i.e., around 10 to 35% decrease depending on the environmental burden factor [53]. It is also apparent that the results often contain technology and management optimization opportunities, the implementation of which ultimately leads to the transformation of consumption patterns [52,54]. When competition cannot take place through cost cutting, product innovation also represents the distinctive, successful factor which can occur through the use of applied creativity, craftsmanship and technological transfer [55]. According to Ferilli et al., debates on the efficacy of culture as a powerful tool of local economic and social revitalization are increasingly frequent, especially in this period of Global economic crisis and obsolescence of traditional growth models and schemes [56]. Sustainable development plays a critical role in supply chain management practices and needs harmonized progress vis-à-vis the economy, social advances, and ecological balance [57,58].

## 4. Material and Methods

The most important elements of culture are language, religion, values, attitudes, customs and different norms of the group or society. When we think about cultural models, we interpret different combinations of these elements. The aim of cultural models is, therefore, to find the elements and evaluate their proportions. Their values are comparable to the cultures studied and conclusions drawn with dominant values [57]. Due to the specific nature of our study, it can be carried out in a qualitative method in the form of an in-depth interview. In-depth interviews provide valuable information for research programs, especially when they compliment other methods of data collection. It should be noted that the general rule for sampling interviews is that the same questions and topics arise from the participants, then a sufficient sample size can be achieved. In this research, we reached the maximum number of respondents because we asked all the players on the supply side.

The process of doing in-depth interviews was divided into two stages (Table 1). In the first phase, we interviewed international students, while in the second phase, we interviewed the heads of local businesses. In the first phase, in-depth interviews were conducted by Masters students of the Szent István University in the framework of Multicultural Management subject. A total of 65 in-depth interviews were conducted in 2018. The length of the interviews was 50 min on average, and the length and language of the transcripts were varied.

In most cases, interviews were conducted in English. Demographic data, where possible, was quantified. The respondents were selected from 22 nationalities, most of whom at the time of the interview named Hungary as their current place of residence. With participants who did not typically stay in Hungary, a Skype or a telephone interview took place. Interviews were recorded, transcribed and translated. The structure of the interview scheme is shown in Appendix A. Our present study does not cover all the topics discussed in the interview; rather, it is limited to what we consider to be important. The SWOT analysis is a well-known, useful tool for providing qualitative information, [59,60]. In the study of Skinner et al. [61], SWOT analysis was used to interpret qualitative and quantitative data Analysis to Inform Healthy Eating and Physical Activity Strategies. The SWOT analysis proved to be a

beneficial tool for incorporating local contextual data and community input into the determination of relevant health promotion strategies. However, no study was found to use SWOT analysis as a framework to show sustainable consumption patterns of international students considering also the supply side from local businesses.

**Table 1.** Structure and purpose of the interview scheme.

| PHASE AND TOPICS | GOALS AND EXPLANATION OF ACTIONS |
|---|---|
| Launch meeting and expert discussion about In-depth interview 1 | Defining the necessary information (possible sources of data), listing the participants, designing the sample, reviewing the ethical rules. (Appendix A) |
| Explore the cultural factors and debut | Presentation of the purpose of the study, informing the participants, providing anonymity, recording |
| Data introduction | Obtaining demographic data (age, nationality, residence, family, friends, leisure) |
| Consumer habits theme | Main food types, a special diet, special foods, the way of eating food, cultural food ceremonies, food role |
| Food choice topic | Influence factors, quality, label, risks, traditional vs. new foods |
| Culture and food consumption | Interpretation of the concept of culture, values, ethnic groups, travel, relationships, cultural knowledge |
| SWOT 1 | What kind of food consumption opportunities does the university ecosystem offer for international students? What are the advantages and disadvantages of sustainable consumption? |
| Explore the changes in the local business premises (In-depth interview 2) | Most popular dishes or foods, Change of the guest circle, Impact of foreigners, Target groups, New trends in food consumption, Change in supply, new trends in restaurant operations (Appendix B) |
| SWOT 2 | The business premises around the university campus (Appendix C) are suitable for serving sustainable consumer needs. |
| Expert meeting and discussion | Evaluating questionnaires, summarizing results and drawing conclusions |

We decided to express our results in an appropriate SWOT-matrix to gain a more comprehensive and transparent view of the participants' opinions. The purpose of applying SWOT method was to explore the factors that have a positive and negative impact on sustainable consumption patterns. Beneficial conditions and exploitable opportunities that can be explored will help to maintain healthy eating habits of international students. Adverse circumstances and critical factors that can be discovered will speed up the loss of sustained consumption patterns. In a SWOT analysis, we ranked the strengths, opportunities, and weaknesses and threats associated with sustainable consumption in a matrix.

In the second phase of the research, in-depth interviews were conducted by experts. All food distribution sites were involved in the study, where international students purchased food on a daily basis during their university stay. They interviewed the head of the unit in two restaurants, two buffets, and a shopping mall. The interviews were conducted at the beginning of April 2019.

With SWOT analysis, we examined the extent to which the business premises around the university are suitable for serving sustainable consumer needs (Table 1).

Evaluation process: A four-member authoring team evaluated in-depth interviews and SWOT analyses. Each researcher examined the transcript of the participant responses and comments, and the results were summarized in the SWOT analysis. The researchers then met and discussed their findings and agreed on the differences between themes and categories. Disagreements were discussed until the four researchers reached full agreement.

## 5. Results

In the first part of this section, we present the descriptive characteristics of the participants in our study. Participants were described by the following demographic characteristics: nationalities, age, gender and current place of residence. In the second part, we introduce the main results of our qualitative research, touching on the main topics of the interview guide.

### 5.1. Characteristics of the Sample

It is important to examine the distribution of participants by nationality if we want to collect information about the cultural background. Participants were members of 22 nationalities. Most came from Germany (7 persons), China (6 persons) and the surrounding countries: Poland (5 persons), Romania (5 persons), and Slovak (5 persons); the rest (37 persons) came from other countries (Figure 1).

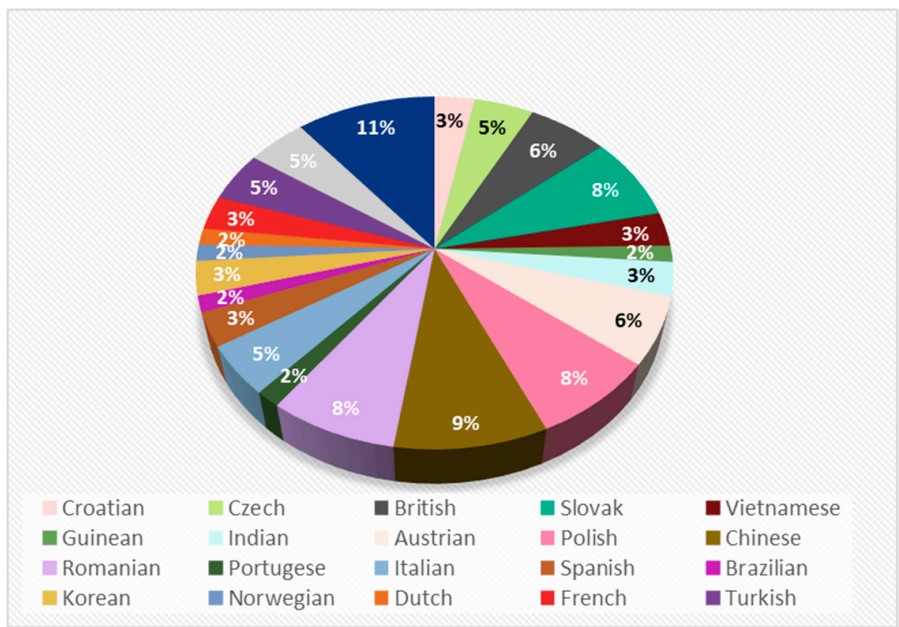

**Figure 1.** The distribution of participants based on nationalities, percent.

Seventy percent of the participants were men, and 30% were women. There were 28 people, aged from 18–29, and 20 from 30–39. Most participants came from these youngest age groups. Older age groups were less represented (Figure 2).

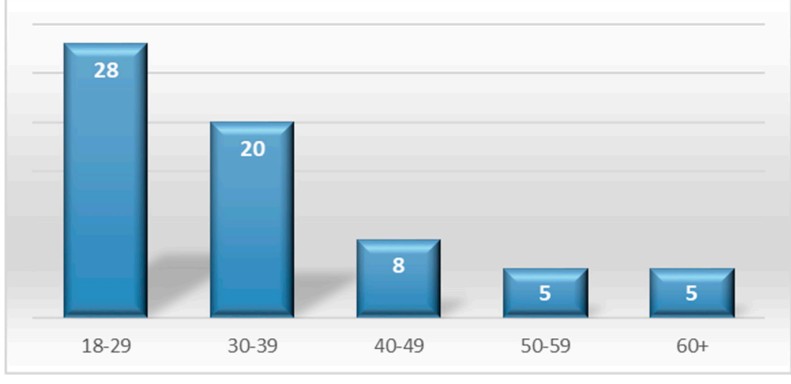

**Figure 2.** The number of participants based on age, persons.

We were also curious about the current location of the participants. The vast majority of participants were living in Hungary at the time of the interview (51 persons). The rest were living elsewhere in

Europe (12 persons): Germany, Great-Britain, Romania, Serbia, Slovakia (each one represented by 2 persons), Austria, Holland, Poland and Spain (each one represented by 1 person). Compared with students who were living in Hungary, the responses of students living elsewhere had much stronger emotions. Many negative memories came up as a result of the questions. Presumably, home conditions have amplified the negative experiences that were worthwhile staying abroad.

*5.2. The Results of Qualitative Research Based on In-Depth Interview 1*

5.2.1. Interpretation of the Word 'Culture' and Sustainable Traditions

Our participants interpreted the notion of culture very differently, but recurring elements can be perceived, such as traditions, habits, and lifestyles. Sustainable nutrition also means the link between different social segments, mainly based on good links between agriculture and other sectors. It is desirable to understand this process because of the poor connection; negative environmental externalities appear on the production and consumption side (structural and qualitative transformation of soil due to the use of pesticides and stimulants, food wastage on the consumer side).

> "Culture is primarily about all of our traditions and customs, including our everyday linguistic expressions, the everyday things that we like to do, the foods we consume, the folk traditions associated with different occasions, and the culture, which is part of people's mentality, what and how do we see it" (Spanish). "It means the traditions and habits of a nation to me: how they live, how they behave, what they do, what their characteristics are in social life and in private, what foods they consume, what their drinks are, what their agenda is, and so on" (English). "For me, culture is a way of life that a group of people follows from generation to generation".
>
> (Guinean)

Some participants believe that culture is a set of sustainable traditions which are handed down from generation to generation.

> "Culture is a set of habits and traditions that this generation produces and delivers to future generations. Among the participants, there were those whose appearance, the look of the culture, is the habit of others, their dressing, the behavior, and the speech, looks".
>
> (Serbian)

Thanks to the strong cultural background, students retain an important part of their sustainable nutritional and social values, even during their extended stay abroad.

5.2.2. Values in Life and Food Consumption

During the interviews, the most frequently mentioned values were family and health, which participants repeatedly linked to food consumption. The consumption of fresh and healthy foods was preferred for the interviewed students. According to the literature, consumers prefer pure, preservative-free foods. The primary reason for this is the protection of health.

> "Of course, children are the greatest gift in our life." Also, love of nature is very important as well as health. "We love to walk and play in nature, so it's important to be healthy" (Austrian). "For me, the key to health is that we try to get the most out of our lives, the more we know the World, the more people we know, the more we get out of our comfort zone, and of course the family also plays an important role in my life" (English). "The family and the health I can point out two of these things: both of my values are related to food consumption since we share the best food with the family, and the foods we eat are very important to health. Fortunately, the most preferred Italian and Spanish cuisine I use is fresh and healthy foods, both of which are based on lightly basic ingredients."
>
> (Spanish)

In addition to family and health, for other people, love, friendship, honesty, truth, and respect are supreme values. "Good human relationships" were also of decisive value for some participants. The answers also showed that students do not precisely know what they call healthy food. In general, they do not have information about environmentally-friendly production systems or climate-friendly products.

5.2.3. Main Food Types and Ingredients Affecting Food Consumption

As to what determinants are taken into consideration when choosing food, the participants mostly considered freshness to be fundamental. The participants also considered the quality, taste, smell, texture, and color of foodstuffs to be important. In the responses of the interviewed students, they clearly indicated that they prefer food from local products because they are fresh and do not contain preservatives and coloring agents. These preferences also confirm European literature, which is related to short transportation and the sale of fresh food. "It is imperative for meals to keep the meat and vegetables fresh. In the case of long-lasting foods, it is important to have as little sugar as possible and not to have any coloring agents or additives." (Italian)

Another critical factor in the selection of foods was the origin, expiry date, and packaging. Participants often choose products during their stay abroad that they already knew. They are similar to home products, although they often differ in taste and packaging. Discovering your favorite products while staying abroad is a long process. According to the respondents, previous experiences are always of great importance, so it is advisable to look for people from the same country or region. This not only allows individuals to share food experiences, but also forms community islands.

The types of food that are popular among participants show a very different picture. However, most people prefer vegetables, meats, and fish. Naturally, almost all participants prefer local, traditional dishes. At the same time, several participants have explained that they are also willing to try other nations' foods (in typical Italian, Chinese, Turkish restaurants).

> "In my culture, people usually consume fish or some Mediterranean food. I also love fish, rice, meats, and vegetables. However, my family eats any food: usually we eat lots of fish, potatoes, vegetables, and pasta" (Italian). "For the older generation in the Netherlands it is essential to eat meat every day: one day pork, next day beef, then chicken. They eat mostly beef and fish. Typically, olive oil is used to prepare meals. The Dutch tradition is raw herring, which is commonly eaten on the market on the spot with onions. Typical food in the local market is the floury egg yolk fried in garlic sautéed with garlic mayonnaise sauce. Fried potatoes are often eaten with special spicy ketchup, curry ketchup. Mustard is mainly consumed for sausage and pickling. The sausage is made from pork meat" (Dutch). "In English culture, dinner is traditionally the main meal, during the day only people eat sandwiches or something simpler, which does not take much time. Nowadays, there are no old habits. Everyone is adjusting their meals according to their taste and routine. My family has continually been a supporter of healthy eating, so we have a lot of vegetables and fruit on our table, and we often eat fish" (English). "In Barcelona, we follow an average Spanish diet. I would mostly describe this as eating many fruits and vegetables. However, we consume a lot of fish as it is effortless to access local markets and is also relatively cheap. There is always a little fresh fish available. Thanks to the Catalan climate, which usually covers sweltering summers and mild winters, we usually prepare light meals. I like typical Catalan dishes such as "pan con tomate" or "arroz negro".
>
> (Spanish)

Favorite vegetables or seasonal foods are not easy to find in Hungarian shops. Availability may vary according to temperature and season, so fresh seasonal vegetables and fruits can be purchased in different months. Fresh seafood is missing in the Hungarian shops, so is necessary to find a compromise for the nations that prefer seafood and want to eat healthily.

### 5.2.4. Eating Habits in Different Ethnic Groups

In many participants (especially from the Far East and Africa), it can be observed that the quantity and quality of food also express social affiliation:

> "Food plays a crucial role in our culture because it tells us what kind of ethnicity somebody belongs to. On the occasion of holidays, women produce more quantity as they offer food in large pots, people sit around and eat with bare hands" (Guinean). "The lack of food is a sign of poverty, and this is not a pleasure for the Vietnamese" (Vietnamese). "Meals are a key element of Chinese culture and everyday life. I'm Chinese, so I grew up and ate authentic Chinese food. My wife is Manju, but there is no significant difference in the flavours of my family and my recipes. This similarity may be since we are both from North and cultural differences typically come from the North-South divide and have different cultures in certain minority areas such as Uyghur or Sichuan".
>
> (Chinese)

According to the respondents, European eating habits differ significantly from those of Asia or Africa. The average time for eating meals is different. People in Hungary often eat alone and do not spend much time eating. In Asian or African cultures, a healthy meal lasts longer and more people eat together. During the interviews, we also studied the participants' experiences in other countries and the differences they found between their own culture and the food consumption patterns of other nations. In terms of results, we received very different but interesting opinions. Different cultures have various ways of eating, cooking and serving food.

> "People's habits are usually determined by their culture because it is their way of living their daily lives, what they eat, what religion they follow, and how they behave in certain situations. For example, those who shout at a conversation, this was a strange thing to me. A striking difference is that more people in Africa still eat with bare hands from one pot and in Europe everyone is eating with separate cutlery from their own plate" (Guinean). "Vietnamese people like to eat in large groups and talk for a long time. It's not customary in the Hungarian culture" (Vietnamese). "Well, as we talk about foods, the cuisine is certainly part of a culture, like many other things, such as language, humour, and the way people usually approach the culture of life. We Italians are, for example, much more open, while you are more serious" (Italian)! "In southern countries, such as Italy, Greece, people are curious about life: they are not in a hurry, they live and enjoy life" (Polish). "People in France are eating very slowly, and Italian people are having dinner late in the evening".
>
> (Korean)

Every nation has its habits that can be observed in everyday life, in private, and in the workplace. The surveyed students not only observed Hungarian eating habits during their university life, but also gathered information about the habits of other international students. If they were found to be useful, they were used afterwards.

> "I do not consider myself a very good observer, but these are usually quite clear and obvious: let's take the British and the Dutch, we have regularity in everything, we love routines, we are not too personal, although it depends on personality, but with the Dutch, who are loose, direct and more flexible" (English in the Netherlands)."In terms of food, I think it is quite natural that every ethnic group uses the basic foods that are easy to reach in the country. In Spain, due to the proximity of the sea, we eat lots of fish, and in addition to the long, sunny summer and mild winter, they've been eating all over the year, and I've learned so far that they prefer locally grown foods in other cultures and foods that are easy to buy" (Spanish). "For people coming from Western civilizations, for example, it is strange that we Chinese are all eating from the same bowls. In our opinion, eating is a more community experience than

in the West and alcohol consumption is typically less. Although there is Chinese beer that we consume at the table, however, it should only be consumed with moderation".

(Chinese)

In Hungary, alcohol is not typical for everyday meals. Other cultures consume naturally smaller amounts of alcohol during lunch. Thus, it is also part of a healthy dietary intake.

*5.3. SWOT Analysis of Sustainable Eating Habits*

In the SWOT analysis, we examine the extent to which international students insist on their home consumption habits. What are the options for keeping their home eating habits? What are the advantages, disadvantages, and opportunities for sustainable consumption during their stay in Hungary? The results are shown in Figure 3. This figure represents Strengths, Opportunities, Weaknesses, and Threats which are described below.

**Figure 3.** SWOT-matrix of sustainable food consumption of participants.

Participants of the study are from many places, and their personal decisions and habits are kept differently. Can we keep our sustainable eating values while abroad? Absolutely yes. But sometimes it is tricky, because we must face with differences in cultural eating habits, and they may diverge from ours to too great an extent. Openness can be critical in this case. Once one goes abroad, she/he takes the time and money to learn the new culture, to explore her/his eating habits in vastly different circumstances and surroundings. It is essential to mention that the references to food consumption preferences or sustainability criteria (carbon footprint, bioproduction, meat or vegetable, water footprint, etc.) specified in the literature were not mentioned or only tangibly mentioned during the interviews. However, the consumption of local foods has appeared almost everywhere during the conversation as the key to healthy eating, and this is an interesting phenomenon, because the participants described the circumstances of their stay abroad.

*Strengths*: It is positive that our participants travel a lot, have considerable experience of the world, are culturally open to nationalities, and they are happy to try local dishes. There are many advantages of buying local foods: local food has less distance to travel, so it can be delivered and sold quickly after it is picked. With local food, one can support the local economy: local food can support hundreds of jobs at shops, even in small towns, and this, in turn, supports producers locally. According to our

results, one of the main influencing factors in consumer food choices is freshness. By buying local food, participants can always rely on the freshness and tastiness of the product. Air-freighting of fruit and vegetables is an essential contributor to greenhouse gas pollution from our food supply system. It is also better to buy seasonally, and think about eating less, but better quality, meat and dairy, to cut the carbon footprint further. Local food may not always be the cheapest food available, but it is still affordable for many, and is generally of good quality. It is also worth buying local food because of the freshness, taste, and quality of ingredients. Processed foods often use cheaper ingredients, making them cheaper but less nutritious.

Many of the participants mentioned that they are trying to preserve their national traditions in a foreign country. We can find a discussion of the importance of traditional food in the study of Balogh et al. [62], where traditional food products can command a substantial price premium compared against mainstream alternative products. In cases where their traditional foods are not available, the participants try to replace their usual foods with locally-available food.

"There are great differences between European and Asian customs, architectural style, dressing, rules, etc. They mainly cook with other ingredients and spices due to its geographical location and climate. I rarely make Thai food, but I use ingredients, spices (coriander, basil, ginger, tamarind) to make our national foods more colorful."

(Slovak in Thailand)

"As you know, our religion is Muslim, so that is why we do not eat pork. Instead of this kind of meat, we eat chicken, poultry, and lamb. We eat a lot of vegetables, for example, paprika, aubergine, beans, lens, and chickpea. We prefer to have for every single food some bread, and usually, have some cheese or marmalade on it " . . . I can keep up with our religious habits in Hungary, as in Hungary halal meat can be obtained from several places, of which very delicious paprika chicken can be made".

(Turkish)

"I think traditional food is the basis of the culture of a nation. I like the traditional dishes of both nations. Traditions must be nurtured and kept alive. "... "I like to try all the new things, food as well. Here I see a small problem with the (mostly older) Hungarians because they insist too much on traditional foods (e.g., white bread). My favorite Czech food is vepřo-knedlo-zélo, that is, fried pork with steamed dumplings (knedlí) and steamed sour cabbage. From the Hungarian dishes, stew (especially beef and mutton) and paprika chicken. I try to grow some vegetables and fruits in my garden, and I try to use organic farming methods."

(Czech)

"I got married in Hungary 30 years ago. It was interesting that when I moved here and started working, I had the opportunity to eat in the canteen. But then I didn't know Hungarian so I tasted all the food and described the name of the food to myself and whether I liked it and I will love it next time or not. During the holidays I introduced Polish dishes at home, for example, I made a feast without meat at Christmas. At the same time, according to Hungarian tradition, we often have fish soup at our holidays. Like in Poland, I often make mushroom and cabbage pierogi. It may also be interesting that there is always herring with boiled potatoes and green onions."

(Polish)

Locality is the main factor that can influence the participants in their food choices. According to the qualitative study, they prefer locally, easily available or locally made, fresh products, contributing to sustainable consumption.

"As far as food is concerned, I think it is quite natural that all nations use the basic foods that are easily accessible in the country. In Spain, because of the proximity of the sea, we eat a lot of fish, and also, thanks to the long, sunny summer and a mild winter, many fresh fruits are available throughout the year. So far, I have found that locally produced and easy-to-obtain foods are favored in other cultures as well" ... "We do not put much emphasis on buying organic food specifically. Fruit, vegetables, and fish are all freshly put on the market and of good quality. I don't feel the need to buy special foods. It is also true that we often buy in local markets because we always have the right quality of food. I buy meat from the butcher and fish in the fish market. I try to produce some of the herbs as I have a small herb garden on my terrace.".

(Italian participant in Spain)

"I prefer light meals, but on a busy weekday, I also enjoy "street food" dishes, let us say, gyros or sandwich on the street. I typically consume a lot of vegetables. I like, for example, asparagus, mushrooms, and peppers prepared mainly for tapas, combined with some fish or shellfish. I use many fresh spices to prepare meals, especially basil and oregano. I also like salads, but there's always some meat on the table beside the salads. I also like Italian pasta specialties."

(Spanish)

"My family loves shopping and loves the markets. We mainly buy from local producers. It is important to me that the food should be local if it is a fresh vegetable or fruit, but it is the same with meat and dairy products. It is a fantastic experience buying an egg from a woman who feeds hens; this creates a personal relationship. Man gets a good quality product and even feels that he has done good to someone. I don't think it's right to eat tomatoes produced in tents, in the winter without sunlight. It was better when we were happy with the tomato when its season came. We only ate peaches when it was ripe. And what is it today? There is always everything, and there is no system in our lives, not in our meal either. This is a disruption to the functioning of our body."

(German)

There are usually plenty of seasonal foods available at a reasonable price. Fruits and vegetables in season are usually field grown, which minimizes their energy demand and carbon footprint. Some people also find it more pleasant to wait for the fruits in the season rather than to eat less flavorsome and more expensive ones in the rest of the year. Foods that travel from far need to be protected by the packaging; this creates millions of tons of wrapping waste. Local food usually sold through markets, traditional shops is often unpackaged or sold in simple bags. By buying local food, we can help smaller farms to survive in the increasing competition. Local food might also be organic, which can help look after the soil. Local producers know their product better. In this way, it is also possible to find out more about the food. Traditions are an important part of the culture. We can conclude from the interviews that most of our participants have strong cultural traditions. Strong cultural traditions are reflected in the participants' values and habits: they have similar food patterns with the family and their ethnic group. The other main strengths according to our research are that most of our participants belong to the younger generation; therefore, they are more open to trying out new foods, to experimenting with new cooking techniques and to extending their food preparation knowledge. A shift in food choice and cooking can promote the development of sustainable food consumption.

*Opportunities*: Many participants prefer traditional local dishes, which is also beneficial because the consumer knows best about the food purchased from the local producer. It can also be a guarantee that environmentally-friendly and ethical food will be put on the table. Not only is it healthier and finer, but the environment can also be protected if the consumer does not choose food delivered from

the other side of the world. Food is not only the source of nutrition but also plays various roles in life. Eating is a social occasion: Food is almost always shared; people eat together; mealtimes are events when the whole family or the village comes along. Food is also an occasion for sharing, for distributing and giving. Many of our participants—especially those from Asian and African countries—explained the importance of the role of food during ceremonies and holidays. In this case, family members usually gather together and eat together from one big bowl. They consume much larger quantities and unique dishes than usual.

*Weaknesses*: Traditions are important in religious and cultural heritage. Strong cultural traditions can also be both a disadvantage and an advantage: too strong cultural traditions may hinder the testing of new foods, the search for new solutions and techniques in cooking. It can be an obstacle to progress, i.e., learning technological advances and new and innovative thinking. Tradition holds one in his/her comfort zone and pushes him/her to continue that old tradition. Another fundamental problem to achieve sustainability is a lack of interest in sustainability. For some of our participants, the environment is an important factor when choosing food. These consumers try not to waste any food; they assume that they only buy such an amount that they can consume. Some environmentally-conscious consumers prefer products that are sold in eco-friendly packaging. Some of the participants, however, do not show any concerns for environmental factors and are not willing to make any effort toward sustainable consumption.

*Threats*: When people travel abroad, one major problem they may have to face is the availability of their preferred familiar foods. It takes a while for one to get used to the local traditional dishes. Some of our European participants had to struggle with the unusual local foods while on a trip to Asia. Another problem can arise if somebody follows a special diet. A special diet can contain fewer nutrients than a normal diet. Moreover, some religions have dietary guidelines which need to be observed. Some of our participants mentioned that there are restrictions in their diet because of religious requirements. They expressed the sentiment that when they travel to other countries, sometimes it is harder to identify those foods (lack of information) that meet their dietary restrictions. We can find negative discrimination because of cultural differences if the people are eating in large groups and talking loudly.

## 5.4. The Results of Qualitative Research Based on In-Depth Interview 2

As described in the methodology section, in the second part of the qualitative research, we examine the impact of international students' purchasing and eating habits on food distribution system on Gödöllő campus at Szent István University. We conducted in-depth interviews at 5 service locations and focused on the following areas: Most popular dishes or foods, Changing guest circle, Impact of foreigners, Target groups, New trends in restaurant operations. Further parts of the questionnaire can be found in Appendix A Figure A1.

### 5.4.1. Most Popular Dishes or Foods among the Students

One of our research questions focused on what are the most popular foods on campus. According to the unanimous opinion of the participants, the best-sellers are sandwiches, especially in vegetable and vegetarian versions. Fruit consumption (e.g., banana, apple) is outstandingly high, chicken dishes are preferred to pork, and internationally known foods include pasta, pizza and fish and chips.

It is clear that healthy food is gaining popularity among consumers. For example, a separate shelf has been arranged in the dormitory buffet for oat-containing foods; another good example of this is the creation of the so-called 'health corner' in the COOP store, where you can see the food labelled in English for the foreign students.

Chicken meat dishes are the most popular on restaurants' menus. Vegetarian foods show a growing trend. Nowadays, a broader selection has to be offered to customers. Previously, 2–3 kinds of food, and now 4–5 kinds of food, are placed on the shelves. The fitness menu is liked very much and is offered primarily with fish and chicken dishes. Demand for special foods is increasing: lactose and glucose sensitivity are the most common. Demand for green salads has changed dramatically

over the past period, with a 300% increase. "Traditional Hungarian food can no longer be sold today, for example, the pork menu (sausage, blood pudding, stew) does not go at all!" (Manager 1). The three-course economical menu is popular, primarily because of its low price. In the case of the food store, the consumption of fruit is very significant, and chicken meat is sold in extremely high quantities. The purchase of semi-prepared meals and sandwiches has increased significantly. Sandwiches are mostly consumed in vegetable and vegetarian versions. The most popular fruits are bananas, apples, lemons.

### 5.4.2. Change of Guest Circle, Much More Foreign Buyers

Looking at the development of the guest circle, we can say that the number of foreign customers is increasing year by year, and it is doubling. "There is a tremendous change in the circle of customers. More than 50% of customers are now foreign students. The majority of students come from Asia and Arab countries. There is also a significant number of Mexican and Spanish-speaking customers. They are looking for domestic flavors in the shop" (Assistant manager). Shops have to react to this. In the case of the university buffet, there is a very large change in the number of customers. Approximately 50% of customers are now international students. International students prefer chicken or meat-free products. The demand for higher-priced products has risen with the appearance of foreigners. Chicken extra sandwiches (+vegetables) and 100% fruit juices are considered to be higher priced products. At the same time, they can be called healthier foods than other products.

> "With the appearance of foreign students, the food supply has changed, which also influenced the consumption habits of Hungarian consumers. Hungarians also choose vegetables and light fitness dishes more often."
>
> (Buffet manager)

Based on the interviews, we can say that the stores have changed in recent years. International students buy much more meat-free and healthy food than Hungarian students. The healthier eating habits of international students also affects the consumption habits of Hungarian students.

### 5.4.3. Impact of Foreigners, Preferences of International Students

The interviews reveal that the consumption of oat products is outstanding in all units and that the use of smack soup is continuously increasing. Chicken meat dominates in meat consumption, in this case, non-breaded chicken, especially without flavoring, is the most popular. Plain, unflavored yogurt and vegetables are consumed with fruit. "The majority of foreign buyers are looking for vegetarian and vegetable products. Of the fleshy products, chicken meat and chicken products are almost exclusively sold to foreigners. Fatty pork is not popular." (Buffet manager).

Foreigners prefer healthier foods such as oatmeal, oatmeal or porridge primarily. Among the sandwiches, they buy chicken sandwiches; Hungarians also have sandwiches with salami and ham. Foreigners also love plain yogurt. But more fruit is consumed, especially bananas. They buy apples and oranges a little less often. International students are looking for familiar tastes from home; what they know. They also buy sweets they know from home, such as Snickers or Twix. Customers prefer international flavors. Foreigners prefer the flavors they consume using at home.

> "There, foreigners almost always order traditional fried chicken with fries, because they only know this. Hungarian flavored foods are not bought. But if, for example, they would offer rice or Chinese food to the Chinese, they would probably not buy it, they would rather laugh because we can't make them suit their tastes. It works the same as when our Hungarians order goulash soup abroad: it just does not resemble the taste of goulash soup in Hungary. But fried foods are usually known everywhere in the world, so they are also welcome by foreigners."
>
> (Manager 2)

The majority of foreign buyers are looking for vegetarian and vegetable products. Of the fleshy products, chicken meat and chicken products are almost exclusively sold to foreigners. Fatty pork is not popular. Smack soups are consumed exclusively by foreigners, mostly Asian students. Coffee habits are completely different in the case of foreigners, they only drink long black, but several times a day. Among the soft drinks, the demand for healthier, larger fruit drinks has increased with the appearance of international students.

### 5.4.4. Target Group, Whom to Sell To

First and foremost, students are the target groups of the shops; everybody spends more year by year, i.e., the average basket price increased from 1500 HUF to 2000 HUF. International students usually spend more on shopping than Hungarian students. Hungarians and foreigners are mixed in the shops.

"55–60% of buyers are Hungarian, 40–45% are foreign. Within this, the distribution is 50% Arabic, 30% Asian, and 20% other. 30% of all products constitute premium one, 90% of which are consumed by foreigners." (Assistant manager).

The guests of the university buffet have changed dramatically in recent years. Among them, 55–60% are Hungarian, 40–45% foreign. Within this, the distribution is 50% Arabic, 30% Asian, and 20% other. In the proportion of all products, 30% buy premium products, 90% of which is consumed by foreigners. 80% of customers are between 18 and 25 years of age, and the 25+ age group is about 20% of the total. The target group for the sale of food store is primarily the younger generation. Students represent 70% of the customers, and the older age group (over 25) is 20%. Local inhabitants make up 20% of the buyers, purchasing all types of products. Students buy a quick-to-eat product primarily. They also eat in front of the shop, for which they buy sandwiches and salads. International students often eat in the public area in front of the shop. The average basket price for two years was 4–5 EUR; now it is 8–10 EUR.

### 5.4.5. New Trends in Food Consumption

Foreigners are looking for familiar flavors and internationally known dishes. The range of healthy foods needs to be continuously expanded to meet the expectations of foreigners. Consumption of traditional Hungarian food is steadily decreasing and is often wholly excluded from the selection. Demand for premium products, i.e., more expensive products, is constantly increasing. These products expand the range of healthier products. International students are very fond of their home consumption habits, looking for products similar to domestic flavors in the local shops. These products usually contain much more vegetables and fewer spices than previously sold foods. In general, food consumption is shifted towards healthier foods. Foreigners buy more healthy foods, but this also affects the consumption habits of Hungarian students.

> "The demand for fitness foods is growing exponentially. This is basically a chicken meat and vegetable meal combination or a fitness meal. The consumption of fried or stacked vegetables is constantly increasing. Breaded cauliflower, mushrooms, zucchini, eggplant are always on offer. Rice is consumed primarily with these vegetables by the customers. Foreigners are looking for fat and low-salt foods, and their consumption of lactose and gluten-free foods is constantly increasing."

(Manager 1)

The university buffet has started to offer to the customer the 'Salad box,' which is made with fresh salad mix and vegetarian and chicken. The demand for gluten- and lactose-free products has greatly increased in recent years. Customers are also requested to order vegan diets. Muslim students ask for chicken diets with special vegetables. International students have changed the product list significantly, but healthier products have come onto the shelves with this change. As the proportion of vegetables and chicken products increases, Hungarian students often choose this instead of pork. The demand for gluten-free and lactose-free muesli products, and for chips has risen sharply in recent years.

The demand for fitness foods is growing exponentially. This is a chicken meat and vegetable meal combination or a fitness meal. Breaded cauliflower, mushrooms, zucchini and eggplant are always on offer. Rice is consumed primarily with these vegetables by the customers. Foreigners are looking for low-fat foods, which, along with lactose and gluten-free foods, are continually increasing. The seasoning of foods is getting lower, using less salt, pepper, and paprika in the kitchen. The taste of food is becoming more and more neutral, more international; everyone can consume it.

### 5.4.6. Change in Supply, More Bio and Healthy Products Come Up

In the case of the interviewed shops, the sources of supply can respond flexibly to new consumer needs. They can obtain any product (bio, natural) or from any country even within days. Due to the increase in the consumption of canned and semi-finished products, the share of imported products in the stock/range also increased. The introduction of innovative products and foods in the shops is continuous. The number of foods on offer in the permanent menu is continuously growing, for example, one of the restaurants offers seven types of menus. Due to the preference for vegetable products, the amount of vegetable delivery in each store has increased. Due to the high level of vegetable consumption, not only seasonal vegetables but also imports of vegetable products have increased.

University buffet suppliers respond flexibly to the changing needs. There are innovative products that manufacturers are trying to compile and distribute according to the needs of foreign consumers. For example, pizza flavored with bacon or so-called global flavors can be found in all product groups.

> "Vegetable (grilled vegetable) fillings and vegetable flavors have become very popular in recent years. The consumption of semi-finished foods has increased by 300% in recent years, mainly focusing on the consumption of oatmeal and smack soups. These are mainly imported products!"

> (Shop assistant)

Restaurant suppliers respond flexibly to changing demand. Most processed products originate from Hungary. Fish species is primarily imported. Canned and braked products are not typical in the supply, but the use of frozen products is regular of the kitchen outside the season. Almost any international product can be ordered through the supplier system. The raw materials from domestic cultivation are preferred by the kitchen, although in many cases, this comes at a higher price. The volume of imported products, whether fresh or durable, is constantly increasing.

### 5.5. SWOT Analysis of the Sustainable Business Premises around the University Campus

According to the in-depth interviews, in the SWOT analysis, we investigated how the food services on the university campus can adapt to the changing demand, and how they can help international students to practice their sustainable eating habits. During the research, we examined the strengths and weaknesses of these providers in terms of following up on changes. The results are shown in Figure 3. This figure represents Strengths, Opportunities, Weaknesses, and Threats, which are described below.

The main question of the SWOT analysis is that the business premises around the university campus (Appendix C) are suitable for serving sustainable consumer needs.

*Strengths*: Based on responses to in-depth interviews, most service providers responded flexibly to changing demand. Special product solutions have been developed in the product segments for which demand has emerged in recent years. For example, this occurred with oat-based products or natural and bio-products. At the request of international students, a fitness menu was introduced in the university restaurant. Simplified and inexpensive menus have been introduced because of a more understandable food supply. A wide range of international flavors and dishes are now available in the food service system, which had not previously been offered. The demand for premium products made from healthy ingredients has increased in recent years, and these have been flexibly handled by business premises (Figure 4).

**Figure 4.** SWOT-matrix of sustainable business premises of uni campus.

*Weaknesses*: There are still many products that international students are not buying from restaurants and shops. Traditional Hungarian dishes are not popular among foreigners because they are usually fatty and spicy. Replacing or transforming them is a great challenge for restaurants. Different age groups present a great challenge in creating a restaurant offering, because a variety of needs have to be matched. The older age group does not consume the same food as the younger international age group. In restaurants and buffets, longer opening hours are required due to different eating habits. Simplified and accelerated payment methods are required in shops.

*Opportunities:* Strong growth in international student numbers is an important business opportunity for campus food service providers. By introducing the right business strategies, sales can be multiplied in the coming years. International students prefer healthy and higher quality foods. Continuous growth in sales and demand side supply can be followed by expanding product range. The introduction of international dishes and the enhancement of food quality can improve the quality of service. Digital follow-up of the menu line, digitization of table reservations and the launch of delivery services (e.g., room service) present further great opportunities in university campus food service systems.

*Threats:* too rapid change in the number of students increases or decreases the flexibility of shops. The disproportionate shift between nationalities also causes inflexibility in the product range. The danger of developing bad trends may be threatened by the preferences of foods that only affect a narrow range of consumers. When designing the food supply, we should try to favor local providing systems. Cheaper and better-suited products are often not available on the local market. Due to the special demand of international students, the volume of imported products can increase significantly.

## 6. Discussion

In our research, we studied the cultural differences inherent in food consumption through in-depth interviews recorded by university students. In our study, we saw that participants interpret the concept of the culture very differently, but we also see recurring elements in the interpretation of the term, such as tradition, customs, and lifestyle. There is also a large variety of foods eaten. It was evident that foods play a vital role in the daily lives of individual nations. Foods and meals, however, serve not only for the living; they also fulfil other functions in the given society, for example, food expression may be part of a social class, reveal income conditions, and express identity. This is wholly consistent with what was found in our literature review. It should be highlighted, however, that our study was devoted

to exploratory research, whose primary purpose was to get to know the cultural factors affecting food consumption. We also wanted to highlight the relationship between the type of food consumed and sustainability. In the case of international students surveyed, the meaning of sustainable food consumption is primarily related to the family, community life and the preservation of health. Based on the survey, we can conclude that the sustainability food characteristics defined in the scientific literature do not appear among the personal preferences. There is an active collaboration between people living abroad in the food consumption system. International students regularly eat in restaurants where they can get closer to their cultural background. The specific impact of food consumption on community life is best seen where the size of the population belonging to a given nation increases.

The results of the first phase of the research have shown that international students retain their home consumption habits abroad. They are looking for meals in restaurants that resemble home flavors and are familiar with their own culture. They buy products in stores that are very similar to or the same as those in stores in their home countries, from which we can say that for the traditional dishes which they cannot find in restaurants, they will use the local market in order to cook their preferred foods by themselves. Our results are in line with different studies who proclaimed that dietary habits acquired in childhood tend to be maintained into adulthood [63,64].

Considering the study of Gerbens-Leenes and Nonhebel [65], which reported that even small changes in food-consumption patterns can trigger large impacts on ecosystems due to the agricultural area required, it may be stated that changing consumption patterns from non-meat-dominant to meat-dominant patterns in many countries will lead to high pressure on water resources required to produce those products [66]. We can say that most of the interviewed students identify a sustainable diet by consuming the usual food at home. It can also be stated that the consumption of healthy foods during stays abroad is much more significant than at home. This is an interesting phenomenon, but it is understandable that they are paying more attention to healthy eating in an unknown place. Increasing consumption of healthy foods like chicken meat and vegetables instead of other types of meat (pork, beef) would have advantages for a sustainable development. It turned out that international students can follow their consumption habits at home if nearby shops and food service providers respond flexibly to changes in demand.

However, foreign consumers also like the typical, fresh and unprotected foods, but these are not always available in local shops. These are fish and seasonal vegetables in our study. Higher demand for special goods may increase the proportion of imported products, which is a negative development in terms of sustainable consumption.

The research revealed interesting relationships between sustainable agricultural production/ greenhouse gas emissions and consumer behavior. There was no opinion among consumer preferences that chose food for the effects of climate change. We have not found any answers to preferences for the local product instead of the global product due to water saving or reduced energy consumption. However, from a sustainability point of view, consumer responses were correct, perhaps due to cultural habits, consumers prefers healthy, low-carbon and local foods.

The results of this research provide a good basis for planning future quantitative research to measure different migration effects and to develop appropriate models that depend on the proportion of different nationalities. From the point of view of sustainable consumption, it is very important that the target group have information on the subject. It is relevant to know what knowledge decisions are based on. Changes in local supply systems can also have a significant impact on the relationship between consumption and local production. In order to avoid these effects having unexpected and unfavorable effects, appropriate conclusions could be drawn from further studies.

## 7. Conclusions

One of the most important findings of the research is that the continuation of studies on food consumption habits and community behavior is crucial, because students are interested in knowing about the diet and cultural traditions. They are happy to read information on the food adaptation habits

of other nations, and to develop their consumption habits and community preferences. The study proved that eating habits in Hungary have an impact on the eating habits of international students, changing them in several respects. Based on the study, it can be concluded that the decisions related to meals are complex, having a significant environmental and social impact. However, it must also be said that thanks to strong cultural backgrounds, students retain a meaningful part of their nutritional and social values, even during more extended stays abroad. The results of the research show that it is challenging to determine the real food needs of a given community. Aspects of sustainable or healthy food consumption often change, and depend to a large extent on the ethnic composition of a given community. So, the question is: how do cultural influences affect these processes in global systems? The result of the research is that cultural influences have an impact on the consumer environment. In the case of international students in Hungary, healthy food like chicken meat and vegetables is dominant, which is a good sign for sustainable development. However, we cannot give an accurate answer to the question of how long people keep their birthplace consumption habits abroad. Nor do we know exactly how the food supply systems of the target areas change as a result of the appearance of foreign consumers, but the fact is that they change. Different consumption needs, which are steadily increasing with the appearance of foreigners, result in consumer confusion and experts competing to win consumers. Chaos is almost complete in a sustainable diet. We cannot decide today what the useful food ingredients for our body are. Also, based on birthplace or cultural background, we call a diet sustainable or untraceable. These aspects, depending on the strength of their appearance, can strongly influence the supply chains or the related producer background. The majority of the respondents also said they prefer buying and consuming local products because they are fresh, and therefore, safer. Local foods are usually cheaper, and consumers would like to support local producers by buying local products. The impact of foreigners' consumer habits on local food chains is, therefore, very complex, as they seek different content based on their cultural habits. As a result, the former customary food supply is constantly being transformed, also affecting producer and supplier systems through the purchase of local and inexpensive products. How these effects affect the supply chain depends largely on the proportion and number of foreign consumers. The results of the research can help marketers better understand consumer behavior and the differences between nations, and secondly, provide the basis for later representative, quantitative research.

## 8. Limitations

The research was done on a small and unrepresentative sample, so no trends can be inferred from the composition of the respondents. Our study does not cover all the topics discussed in the interview, but is limited to what we consider to be important. The presented research does not represent a typical collection and cultural identification of international people who live in Hungary. In the case of in-depth interviews, generalizations of the results were not possible because small samples were chosen and random sampling methods were not used. The purpose of this study was therefore only to determine the direction and nature of further quantitative research. However, the study has produced new results that have not been published so far, and scientific conclusions can be drawn therefrom.

**Author Contributions:** Conceptualization, C.F.; methodology, N.N. and I.R.; formal analysis, P.Y.; resources, N.N.; writing—original draft preparation, N.N. and I.R.; writing—review and editing, P.Y.; supervision, C.F.; project administration, P.Y.; funding acquisition, C.F.

**Funding:** This research received no external funding.

**Acknowledgments:** Preparation the manuscript and our final article was supported by the Climate Change Research Centre and Doctoral School of Management and Business Administration at Szent István University.

**Conflicts of Interest:** The authors declare no conflict of interest.

## Appendix A

---

### INTERVIEW GUIDE

### EXAMINATION OF THE CULTURAL ASPECTS OF FOOD CONSUMPTION

**I.   THE AIM OF RESEARCH (5 minutes)**

1. **Greetings and introduction**
2. **The aim of research:** The aim of the research is to explore the eating habits of different cultures: what kind of differences can be explored in the eating habits of different nations. What kind of factors influence food choice, especially cultural characteristics, such as values, family, tradition, religion, sustainability.)
3. **GENERAL RULE:** There are no no good or bad answers.
4. **Anonimity, recording**
5. **Introduction of the main topics of the interview**

**II.   INTRODUCTION (10 minutes)**

- Could you please introduce yourself with a few words?
- What is your name? How old are you? What is your profession? Where did you go to school?
- Where were you born? In which country? What is your nationality?
- Where do you live at present? How long have you been living at your current place?
- Could you talk a little bit about your family?
- Who are your friends? How often do you meet them? What do you usually do together in your free time?
- What is your hobby? What do you like doing in your spare time?

**III.   CONSUMER HABITS (20 minutes)**

- What kind of food people usually eat in your culture? What do they prefer? And what about in your family?
- Generally, what type of food do you eat? Are there any food that you don't eat or you are not allowed to eat?
- What are your favourite dishes? Are these healthy or not?
- Do you follow any kind of special diet? If yes, which one and what is the reason?
- How often do you eat in a day?
- Where do you usually eat daytime?
- With whom do you usually eat together?
- Could you please tell me: what does a typical family lunch look like in your home? Have you got any special habits?
- Are there any differences between the food that you eat weekdays and in the weekends?
- Who does the cooking in your family?
- Do you cook? How often do you cook?
- How do you usually prepare the food in your home and in your country?

---

**Figure A1.** *Cont.*

- How do the eating habits change on festive occasions in your culture?
- What role does food play in your culture?
- Besides traditional food, what kind of special food do you eat? (e.g. food supplements, organic food, functional food, sustainable food etc.)

**IV.　THE FACTORS OF FOOD CHOICE (10 minutes)**

- What factors do you take into consideration when choosing food? F.e. Traditional, climate friendly, healty, bio etc.
- What do you usually look at first on the food product when purchasing?
- How important is quality for you? And what about the risks?
- What do you usually read on the label (fe. bio, green, low-carbon) of the product at first?
- Are you afraid of any risks in food consumption?
- Who buys the food in your family? What kind of shops do you prefer when it comes to food purchasing?
- How much do you insist on traditional food?
- How much are you willing to try new or unknown food? Are you ready to cook such kind of food?

**V.　CULTURE AND FOOD CONSUMPTION (15 minutes)**

- What does the definition of culture means to you? Could you please describe your thoughts?
- What are the main values in your life?
- How important these values are in your food consumption?
- From the statements below, please choose the one that you most agree with:
  - ❖ I pay attention to healthy nutrition and balanced diet.
  - ❖ For me it is more important that the food is tasty even if it is not so healthy.
  - ❖ Price is the most important factor in food choice.
  - ❖ The lunch that I consume together with my family is the most important for me.
  - ❖ In my opinion, healthy diet can attribute to the prevention of diseases.
-
- What type of food do you know that originally belongs to other nations? How often do you prepare or buy such kind of food?
- How often do you meet people of different origin, from different countries?
- Do you travel abroad? How often? When did you go last? What was the reason?
- Which was the the longest stay/time that you spent in a foreign country? Why?
- What kind of differences did you meet between the habits of people with different cultures?
- What cultural differences did you notice between the eating habits of other nations and your nation?
- What can be the reasons of these differences?
- Was it hard for you to accomodate yourself to these differents habits when you were abroad? Could you describe your feelings?

**VI.　CLOSING THE INTERVIEW (5 minutes)**

Interviewer, please draw the conclusions. Say thank you for the participation.

**Figure A1.** The structure of the In-depth interview 1 (questions of the interview guide).

**Appendix B**

**Table A1.** The business premises interviewed.

|  | Name of Premises | Type of Premises | Participant | Business Turnover |
|---|---|---|---|---|
| A1. | SZIE Kalória Kft. MENZA Önkiszolgáló | Restaurant | Manager1 | 500–600 customers/day |
| A2. | Menza Bisztró | Restaurant | Manager2 | 700–750 customers/day |
| A3. | SZIE Bagoly Büfé | University Buffet | Buffet Manager | 300–400 customers/day |
| A4. | Koli Büfé | Dormitory Buffet | Shop assistant | 600–700 customers/day |
| A5. | Gödöllő COOP SZIE | Food store | Assistant manager | 1200–1300 customers/day |

**Appendix C**

The structure of the in-depth interview 2 (questions of the interview guide):

Type: Face-to-face in-depth interview; Time: 40–50 min; language: Hungarian, Recorded and anonymous form.

Guided Questions:

– Which dishes are most popular in your restaurant/shop? What types of foods do customers prefer? (pasta, pizza, vegetarian food, street food, salad, alcohol, etc.)
– Have you experienced any changes in food consumption in recent years as a result of the increased presence of foreign customers? If so, what kind of change did you experience? (e.g., rate of meat-vegetable consumption, special needs, etc.)
– What type of guests visits most in your restaurant/shop? (Who make up your target group?) (Nationality, age, wealth, etc.)
– Have you experienced any changes in the guest circle in recent years? (very changed—moderately changed—changed slightly—never changed)
– Are there any new trends in restaurant operations? (e.g., prefer fresh produce, bring home foods, prefer local foods, new healthier foods)
– Does the appearance of foreigners affect your offer? Did you need to change something on it? (introduction of nationality products, expansion of supplier relations)

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
