# Peer review of "The Role of Cultural Factors in Sustainable Food Consumption—An Investigation of the Consumption Habits among International Students in Hungary"

_sustainability, doi:10.3390/su11113052_

Round 1

Reviewer 1 Report

The authors reviewed the paper The influencing role of cultural factors in sustainable food consumption – a short communication paper on the consumption habits of international students in Hungary, and the more appropriate reflection of the research methodology (materials and methods) as well as the research results, determine us to appreciate the efforts of the research team and to propose the paper for publication.

As mentioned in the review report, research methodology is appropriate and reflects cultural models to find the different elements of culture based on the survey method through a qualitative method in the form of an in-depth interview. The elements / values mentioned by the author as the most relevant and most frequent in interviews were family and health. However, it relies more on empirical data and does not reflect research-type elements but only tools (as is the case with the interview). If the authors could demonstrate the econometric model confirming the research results would confirm the academic level of the paper and the research capacity of the team.
We also believe that authors need to reflect more deeply on the causal and sustainability links between food choices and their impact on the environment and society

Author Response

Dear Reviewer, 

Thank you very much for your comments and your suggestions, we really appreciate them! 

Please find attached our answers to your comments related to our paper "The role of cultural factors in sustainable food consumption – an investigation of the consumption habits among international students in Hungary".

Reviewer 2 Report

Please modify the title: The role of cultural factors in sustainable food consumption – an investigation of the consumption habits among international students in Hungary.

Please analyse the impact of previous experiences in food purchases and preparation on the food consumption behaviour of international students during their mobility. Is there a difference between those who bought food in their families and those who started to do it abroad? Is there a difference between those who cooked food in their home country and those who did not?

Please delete Figure 3 and describe the data only in your main text.

Please refer to:

1. Bryła P., The development of organic food market as an element of sustainable development concept implementation, Problemy Ekorozwoju - Problems of Sustainable Development, 2015, vol. 10, no. 1, pp. 79-88.

2. Bryła P., Organic food consumption in Poland: Motives and barriers, Appetite, 2016, Vol. 105, pp. 737-746.

Language issues:

32 – values abroad

48 – global market

71 – and how

87 – include bio

101 – leeks

120 – diseases

122 – It means that

124-125 – this sentence is incomplete (no verb and object)

157-158 – that is influenced by many factors, creating both challenges and opportunities [35].

174 – delete „Appearing newly”

228 – contributes

243 – wherever

268 – the change of which

270 – cost

287 – “respondents” instead of “samples”

296 – delete “were”

323 - “section” instead of “chapter”

331 – other countries

337 – were represented

341 – The rest

345 – interview responded

346 – “emotions” instead of “impulses”

360 – culture, which

363 - agenda is, and

370 – delete “The basic opinion about the culture”

428 – missing in

428 – so it is necessary to find

429 – and want

430 – ethnic groups

433 – belongs to.

452 – with separate

456 – (Italian).

465 – delete “(who are we have a relatively mixed nation)”

466 – Dutch, who

479 – habits.

481 – Hungary?

485 – of the study participants

547 – buy meat

547 – and fish

557 – done good

559 – its season

560 – there

560 – meals either, this

593 – innovative

609 – lack

611 – time.

627 – do you mean perhaps “food labelled in English”?

631 – is liked very much

632 – increasing:

668 – sweets they

670 – flavours they

689 – products constitute premium ones

689 – which are

713 – with these

722 – The demand for gluten-free

755 – SWOT

757 – habits.

758 – Figure 4

759 – Threats, which

760 – on the basis of the interviews

Figure 4 – Hungarian

802 – a large variety

846 – global

866 – small

Interview guide IV – For example, traditional

Interview guide V – can contribute to

1061 – anonymous

1064 – alcohol

Author Response

Dear Reviewer,

Thank you very much for your comments and suggestions related to our paper!

Please find attached our replays to your comments.  

Wish you all the best!

Authors

Reviewer 3 Report

The paper provides a qualitative view of cultural factors in food consumption. However, the topic of interest is extremely narrow only focusing on the consumption habits of international students. Although SWOT analysis is extremely useful in identifying key factors, this may not be enough. A thorough analysis is required.

Additionally, the discussion in the paper is extremely lacking only providing a general view of the authors' results. Consider extracting the key highlights for the results and substantially discuss them individually in each subsections. At the current stage the authors require a whole revamp for the discussion section.

Author Response

(The authors gave the same response as above.)

Round 2

Reviewer 2 Report

Language corrections:

line 123 - Nevertheless, we should not forget about the importance

124 - geenrates eating habits from

168 - have

179 - tuna is also fish, so it cannot be consumed instead of fish

186 - "research studies" instead of "researches"

193 - and, just

260 - achieved - around

266 - delete "2015"

300 - a framework

315 Consumer habits theme - a special diet

332 Figure 1 - Slovak

352 - of culture

477 - habits.

478 - habits?

Figure 4 - in freshness

483-4 - Please correct this sentence

511 - available, the

516 - Slovak

557 - there

557 - either. This

595 - consumers

599 - face is

628 - liked

663 - Foreigners

739 - global

752 - in-depth interviews

756 - below.

769 - Figure 5

771 - Figure 5

Figure 5 - international

818 - themselves

819 - dietary

867 - vegetables, which

944 - Bryła, P. The development of organic food market as an element of sustainable development concept implementation. Probl Ekorozw 2015, 10, 79-88.

Interview guide - Consumer habits - Are there any foods

Interview guide - Consumer habits - eat on weekdays

1087 - visit most your

Author Response

Dear Reviewer,

Thank you for your comments and suggestions!

Please find attached our response in the word document. 

Wish you all the best!

Authors 

Reviewer 3 Report

The paper is now in a better form and now up to standard for publication. However, there is a need to fix some of the components in the paper.

Figure 1 & 3. Consider changing them as a table, I can't really see the point why a bar/pie chart is used here.

There are two Fig 4 - this needs to be changed accordingly.

There is a ! mark on line 477? Is this necessary?

Author Response

Dear Reviewer,

Thank you for your comments and suggestions!

Please find attached our response in the word document. 

Wish you all the best!

Authors 

This manuscript is a resubmission of an earlier submission. The following is a list of the peer review reports and author responses from that submission.

Round 1

Reviewer 1 Report

Theme of the research paper The influencing role of cultural factors in sustainable food consumption - a case study on the consumption habits of international students in Hungary, is relevant and very clear about the role of food consumption in the culture of several ethnic groups and how cultural factors influence eating habits through research methods appropriate to this purpose. Participants involved in the study belonged to different types of nationalities.

Research methodology is appropriate and reflects cultural patterns with the objective of finding the different elements of cultures based on the survey method through a qualitative method in the form of an in-depth interview. The elements / values mentioned by the authors as the most relevant and frequent in the interviews were family and health.

The concepts and bibliography of the specialty are clearly specified and demonstrate good documentation of the authors of the research, moreover citations are clearly mentioned in the paper.

In terms of results, they are adequately presented; more according to interviews, the presentation of the results through a SWOT matrix presents in the authors' view the way people from different nations have insisted on their food, which means that they have a direct effect on their sustainability.

The findings reflect the most important research results and the continuity of nutrition studies, consumption habits and community behavior as a result of the authors' views and research being essential because students are interested in knowing about diet and traditions cultural. This aspect determines us to see the targeted aspect of the authors, ie the work is a bibliographic source for students to learn about food adaptation habits of other nations in order to develop their consumption habits and community preferences. Moreover, the study argues that food habits in Hungary have an impact on the eating habits of students and foreign students changing them into several elements. However, in the conclusions we would like to find the opinion of the scholar researchers on the scientific aspects that have an impact on the individual's behavior regarding the table decision, namely that it is a complex decision that has a significant environmental and social impact.

With this small additional additions to the chapter of conclusions, we propose to publish the paper.

Reviewer 2 Report

The scientific contribution of this manuscript is unclear to me. The relationship between culture and sustainable consumption has not been explored in a sufficient way to justify publication. That is why I recommend rejecting this manuscript.

Moreover, the quality of English is very low. English proofreading will be necessary if the authors decide to revise this manuscript.

Reviewer 3 Report

The paper explores cultural factors in sustainable food consumption in Hungary. This paper however provides limited information on what model that was used and the methodological approach applied in the study. Discussion was also lacking. Further amendments is required for this paper to be accepted for publication.

Introduction

The authors should consider adding research hypothesis in the last section of Introduction.

Methodology

Some expansion on what model that the authors follow is required here, the method seems to be conjured up out of nowhere without any references towards a study? Questions that was used should also be mentioned here.

Results

Figures: Pie chart is somewhat blurry, Figure 3 GB and THE...? Perhaps consider merging the figures to a table as demographic information? Figure 4 is blurry.

Discussion

The discussion provided in this manuscript is not sufficient and should cover more on what the research hypothesis at the moment the discussion seems brief and lacks of impact in highlighting their main results from the SWOT analysis.

Some other minor comments

Line 248 - indent paragraph in?

Line 263 - ,, to " for quotation

Line 278 - what does own edition mean here?